# Body Dysmorphic Disorder in Aesthetic and Reconstructive Plastic Surgery—A Systematic Review and Meta-Analysis

**DOI:** 10.3390/healthcare12131333

**Published:** 2024-07-04

**Authors:** Joseph D. Kaleeny, Jeffrey E. Janis

**Affiliations:** Department of Plastic and Reconstructive Surgery, The Ohio State University Wexner Medical Center, 915 Olentangy River Road Suite 2100, Columbus, OH 43212, USA; joseph.kaleeny@osumc.edu

**Keywords:** body dysmorphic disorder, plastic surgery, prevalence, meta-analysis

## Abstract

(1) Background: Body dysmorphic disorder (BDD) presents significant challenges in aesthetic and reconstructive plastic surgery, impacting patient outcomes and well-being. Understanding its prevalence and associated factors is crucial for effective patient care. (2) Methods: A systematic review of national and international databases on body dysmorphic disorder, plastic surgery, cosmetic surgery, reconstructive surgery, and prevalence yielded 999 studies between 1878 and April 2024. Inclusion criteria focused on studies reporting prevalence while excluding those with small sample sizes (<20 participants), unclear diagnostic criteria for BDD, and non-English accessibility. (3) Results: A meta-analysis using a random effects model was conducted on 65 studies involving 17,107 patients to estimate the prevalence of BDD. The overall estimated prevalence of BDD was 18.6%; 10,776 (62.9%) were females, with a mean age of 35.5 ± 11.7 years. Subgroup meta-analysis found significant variability in effect sizes across countries and types of specialty, of which Brazil showed the highest proportion and dermatology exhibited the smallest. Meta-regression analysis found no significant relationship between the year of publication and prevalence rates. (4) Conclusions: Our findings update the current literature on BDD prevalence in aesthetic and reconstructive plastic surgery. We emphasize the importance of proactive screening and multidisciplinary care approaches to address the complex challenges posed by patients with BDD. Further research is needed to explore evolving trends in BDD prevalence and factors influencing its expression across different cultural contexts.

## 1. Introduction

Body dysmorphic disorder (BDD), characterized by an overwhelming concern over perceived flaws in physical appearance, represents a complex psychiatric condition with significant implications for individuals’ psychological well-being and social functioning [1]. Enrico Morselli, an Italian psychiatrist in 1891, first coined the term ‘dysmorphophobia,’ derived from the Greek word ‘dysmorfia’, meaning ugliness, to describe people who perceive themselves as flawed but have no apparent physical deformities [2,3]. This early recognition laid the foundation for understanding and diagnosing BDD, highlighting the enduring significance of addressing distorted body image perceptions in mental health discourse.

Recently, epidemiologic studies have reported a prevalence of BDD ranging from 0.7% to 2.4% in the general population [4,5,6]. With a considerable population affected, treating and managing patients with BDD presents unique challenges for providers of stigma, diagnostic barriers, treatment resistance, access to specialized care, and long-term recovery [7,8]. In an attempt to address perceived imperfections, patients with BDD will often request cosmetic and reconstructive plastic surgery, the prevalence of which among these individuals is the subject of clinical interest and debate [9,10].

Cosmetic surgeries and procedures in patients with BDD present complex challenges, although historically have been considered a clear contraindication [11,12]. Despite the potential benefits of plastic surgery in addressing physical concerns, individuals with BDD may experience dissatisfaction with surgical outcomes, irrespective of the quality of technical execution or subsequent results, or pursue unnecessary procedures that can exacerbate their psychological distress [13]. This emphasizes the importance of understanding the prevalence of BDD among patients seeking intervention. Recent systematic reviews and meta-analyses have sought to determine the prevalence of patients with BDD in dermatologic and aesthetic settings which have shown rates ranging from 12.65% to 19.2% [14,15,16]. We aim to update these with more recent findings across a more diverse population with a broader range of variables considered, providing comprehensive and valuable insights into the complex relationship of BDD prevalence in plastic surgery.

## 2. Materials and Methods

### 2.1. Literature Search Strategy

A systematic literature search was conducted to identify relevant studies on the prevalence of BDD among patients presenting for cosmetic and reconstructive plastic surgery. We followed the guidelines outlined in the PRISMA statement (Preferred Reporting Items for Systematic Reviews and Meta-Analyses) to ensure transparency and comprehensive reporting. The search was performed in electronic databases, including Cochrane, Embase, ScienceDirect, Scopus, PubMed, Web of Science, PsycINFO, and Google Scholar. The search strategy utilized a combination of Latin keywords related to body dysmorphic disorder, plastic surgery, cosmetic surgery, reconstructive surgery, and prevalence. The search was restricted to studies accessible in English between 1878 and through April 2024.

### 2.2. Study Selection

The search results were screened for eligibility based on predefined inclusion and exclusion criteria. Studies were included if they reported data on the prevalence of BDD among patients with cosmetic or reconstructive plastic surgery, including papers within specialties of dermatology, oral-maxillofacial surgery (OMFS), and otolaryngology (ENT) who perform similar procedures. Exclusion criteria included cases unrelated to the topic, studies with small sample sizes (<20 participants), studies lacking clear diagnostic criteria for BDD, studies with incomplete or missing data, and studies inaccessible in English. In the initial search, 999 articles were identified, of which 336 were duplicates and were removed. A total of 663 were screened by title and abstract, of which 111 were sought for retrieval and entered the qualitative third-phase assessment. Sixty-five records published between 1998 and 2024 were included in the final analysis (Figure 1).

### 2.3. Quality Assessment

To ensure the quality and transparency of our systematic review and meta-analysis, we employed the combined cohort, case-control, and cross-sectional Strengthening the Reporting of Observational Studies in Epidemiology for Cross-Sectional Study (STROBE) checklist [17]. This facilitated the evaluation of the methodological quality of included non-randomized studies by assessing key domains such as selection, comparability, and outcome ascertainment. Articles meeting six to seven criteria were classified as high-quality, while those meeting less than two and between two and five criteria out of the seven were considered medium and low methodological quality articles, respectively [10,15].

### 2.4. Data Extraction

Data were extracted from included studies using a standardized data extraction form. Extracted data included study characteristics (e.g., first author’s name, publication year, country of origin) and participant characteristics (e.g., sample size, mean age, BDD prevalence, and patient sex).

### 2.5. Ethical Considerations

This study involved the analysis of previously published data; no ethical approval was required. All data were retrieved from publicly available sources, and confidentiality of study participants was maintained throughout the analysis.

### 2.6. Data Synthesis and Analysis

A meta-analysis was planned to estimate the pooled prevalence of BDD among patients across included studies, utilizing Stata/BE 18.0 software. Prevalence variance was computed using the binomial distribution variance formula, with the weighted mean employed to aggregate prevalence rates from various studies. A test of homogeneity was performed to assess heterogeneity, and the I^2^ index categorized it into low, moderate, or high levels. Additionally, meta-regression analysis was conducted to explore the correlation between BDD prevalence and study year/sample size. Egger’s test, along with its corresponding funnel plot, was applied to examine publication bias. A leave-one-out meta-analysis was conducted to assess the stability of the pooled effect size estimate. Subgroup meta-analyses by country and type of specialty were performed to explore variability in effect sizes across different subgroups. Bias assessment tests were also performed to indicate small-study effects. Additionally, a trim-and-fill analysis was conducted to assess the impact of publication bias on the observed results.

## 3. Results

A meta-analysis was conducted across 65 studies to estimate the prevalence of BDD in plastic surgery patients. The articles encompassed a total of 17,107 participants (10,776 female, 62.9%), with a mean age of 35.5 ± 11.7 years (Table 1). Table 2 and Forest Plot, Figure 2 present the prevalence estimates, 95% confidence intervals (CI), and weights assigned to each study, calculated using a random-effects model. The overall estimated prevalence of BDD was 18.6% (95% CI: [15.1%, 22.4%]).

The test of homogeneity indicated significant heterogeneity across studies (Q = χ^2^ (64) = 2120.63, <0.0001), supporting the utilization of a random-effects model. The estimated tau^2^ of 0.1369 indicates variability beyond chance. The I^2^ statistic, representing the percentage of total variation across studies due to heterogeneity, was calculated to be 97.27%, further indicating high heterogeneity, also demonstrated by the Galbraith Plot (Figure 3). The 95% prediction interval for the true underlying effect size (invftukey (θ)) was estimated to be [0.153, 1.644]. The test of θ = 0 yielded a statistically significant result (t (65) = 17.18, *p* < 0.0001), suggesting the observed effect size is unlikely to be due to chance alone. 

Sensitivity analysis was conducted to assess the impact of different fixed values for tau (0.75, 0.50, and 0.25) on between-study variance and heterogeneity in the analysis. When tau was fixed at 0.75, a high level of between-study variance (tau^2^ = 0.75) and significant heterogeneity (I^2^ = 99.49%, H^2^ = 196.10) were observed. The effect across studies remained statistically significant (*p* < 0.0001), however. Fixing tau at 0.50 resulted in decreased between-study variance (tau^2^ = 0.50) and reduced heterogeneity (I^2^ = 99.24%, H^2^ = 131.07), while the effect remained significant (*p* < 0.0001). Finally, fixing tau at 0.25 further reduced between-study variance (tau^2^ = 0.25) and led to decreased heterogeneity (I^2^ = 98.49%, H^2^ = 66.03), indicating a trend towards homogeneity. Additionally, proportions of total variance and prediction intervals remained relatively stable, suggesting increased precision in predicting the true effect size in future studies. 

A leave-one-out meta-analysis was performed to evaluate the stability of the pooled effect size estimate. The proportion of the effect size ranged from 0.151 to 0.224 when individual studies were excluded. All changes were statistically significant (*p* < 0.001).

A subgroup meta-analysis investigated the distribution of effect sizes across diverse subgroups delineated by country and type of specialty (Table 3). The proportion of effect sizes displayed significant variability across different countries (*p* < 0.001) alongside substantial observed heterogeneity (I^2^ = 97.27%). Brazil exhibited the highest proportion, with a Freeman–Tukey’s *p*-value of 0.384, whereas Ireland had the lowest at 0.025. Heterogeneity across countries ranged widely from 0% to 97.60%. Within the spectrum of surgical types, dermatology presented the smallest proportion (0.121), contrasting with plastic surgery, which displayed the highest (0.216). Heterogeneity within surgical types demonstrated a range from 0% to 97.47%. Furthermore, tests of group differences unveiled significant variation both across countries (*p* < 0.001) and types of specialty (*p* = 0.022).

A meta-regression analysis was conducted to explore the relationship between the prevalence of BDD, sample size, and the year of study. The findings revealed a significant negative relationship between total sample size and effect sizes (coefficient = −0.00112, *p* < 0.001), suggesting that larger studies tend to yield smaller effect sizes. Conversely, a positive association was observed between the number of events and effect sizes (coefficient = 0.00656, *p* < 0.001), indicating that studies with more events tend to report larger effect sizes. However, no statistically significant relationship was found between the year of publication and effect sizes (*p* = 0.089). The intercept, representing the effect size when all predictors are zero, was not statistically significant (*p* = 0.114), implying no meaningful effect size under those conditions. Despite these predictors, substantial residual heterogeneity remained, suggesting the presence of unaccounted factors influencing effect sizes.

A bias assessment using the Egger regression test to evaluate the presence of small-study effects is demonstrated in Figure 4. The Egger regression test for small-study effects indicates a statistically significant outcome (t = 3.39, *p* = 0.0012), suggesting evidence of small-study effects in the included studies. The estimated coefficient (β1) was found to be 4.30 (SE = 1.267), indicating a bias towards larger effect sizes in smaller studies. A nonparametric trim-and-fill analysis of publication bias was conducted. The observed effect size was found to be 0.898 (95% confidence interval: 0.806 to 0.991). No imputed studies were added during the analysis, suggesting that the observed effect size remained unchanged even after considering potential publication bias.

## 4. Discussion

The complex interplay of genetic, biological, and environmental factors in BDD remains unclear, but individuals often turn to plastic surgery as a solution to alleviate perceived flaws [82]. As societal attitudes toward physical appearance and cosmetic procedures continue to evolve, the demand for plastic and reconstructive surgery surges [83]. Understanding the prevalence of BDD among individuals pursuing such procedures becomes increasingly crucial for healthcare providers, especially plastic surgeons. To explore the prevalence of BDD in this population, our study drew data from 65 studies globally, including over 17,000 individuals. We revealed that the overall estimated prevalence was 18.6%, similar to previously reported rates, and substantially higher than the general population of less than 3%. 

Our study demonstrated significant variability across different specialties and countries. Patients pursuing treatment presenting through dermatology exhibited a relatively smaller prevalence compared to others. In one cross-sectional study across 17 European countries in dermatology outpatient clinics, Schut et al. reported a prevalence of BDD among 5487 patients at 10.5%, similar to our results of 12.1% among 5588 globally [84]. This consistency emphasizes the need for comprehensive screening and management protocols across medical specialties to ensure early detection and appropriate support for individuals affected by BDD. Further, across countries, while the vast majority of studies were published in the United States, Brazil showed a larger effect size on average. These disparities stress the complex relationship of cultural, social, and healthcare system factors in shaping the prevalence and expression of BDD. 

Investigating factors such as the study year, we observed no clear increased trend in BDD prevalence over time. While a potential positive association emerged, it lacked statistical significance. This finding raises intriguing questions about the stability of BDD rates amidst the growing popularity of plastic surgery and cosmetic procedures. One explanation might lie in the increasing societal acceptance and accessibility of cosmetic procedures, driven by advancements in plastic surgery techniques and the influence of social media and celebrity culture [85]. These questions garner depth and resonance in light of a recent video documentary featuring Professor Mark B. Constantian, MD, FACS. He engaged in a profound conversation with a patient grappling with severe BDD, ultimately unveiling a journey towards self-acceptance and healing [86]. He described BDD as a spectrum where the desire for surgery often originates from the perceived significance of imperfections. This fosters dissatisfaction, impacting psychological and social well-being [87]. Dr. Constantian observed the patient’s evolution from broad self-loathing to a specific focus on one body part, akin to examining it through a magnifying glass. Despite encountering post-surgical dissatisfaction, similar to many patients with BDD, the patient demonstrated resilience by actively pursuing self-improvement.

Given the profound impact of BDD on individuals and the rarity to resolve, our study underscores the critical need for preoperative screening and ongoing support mechanisms that can enhance patient outcomes and satisfaction [88]. Although recent studies have sought to identify validated BDD screening tools, like the Body Dysmorphic Disorder Questionnaire (BDDQ) and the Dysmorphic Concern Questionnaire (DCQ), across various specialties related to plastic and dermatologic settings, they have ultimately demonstrated varying criteria, regional preferences, an absence of uniform guidelines and the need for further development based on DSM-V [89,90]. In addition to early detection and timely psychiatric referrals, holistic management of BDD necessitates ongoing multidisciplinary support, empathetic counseling, setting realistic expectations, and therapeutic options such as selective serotonin reuptake inhibitors and cognitive behavioral therapy [91]. Furthermore, longitudinal studies tracking changes in BDD prevalence over time and across different cultural contexts can offer insight into evolving trends and patterns in body image perceptions, guiding more effective interventions and support strategies.

### Limitations

Several limitations arose which included potential sampling bias due to missed or excluded studies, generalizability, and publication bias skewing prevalence estimates. Language and publication biases may arise from the focus on English publications. Various assessment tools exist for diagnosing BDD, potentially leading to inconsistencies in prevalence estimates across studies. Tools such as the BDDQ and the DCQ are commonly used but may vary in sensitivity and specificity, influencing the detection of BDD symptoms in study populations. Despite efforts to assess study quality using the STROBE checklist, variations in reporting standards remain. 

## 5. Conclusions

Our meta-analysis revealed a significant prevalence of BDD in plastic and reconstructive surgery, estimated at 18.6% across 65 studies and 17,000 patients. While no clear increasing trend in BDD prevalence over time was identified, further research is warranted to explore evolving trends in body image perceptions. Overall, our study emphasizes the importance of proactive interventions and collaborative efforts to improve patient care and outcomes. The findings underscore the need for standardized screening protocols and multidisciplinary care approaches to address the complex challenges posed by BDD.

## Figures and Tables

**Figure 1 healthcare-12-01333-f001:**
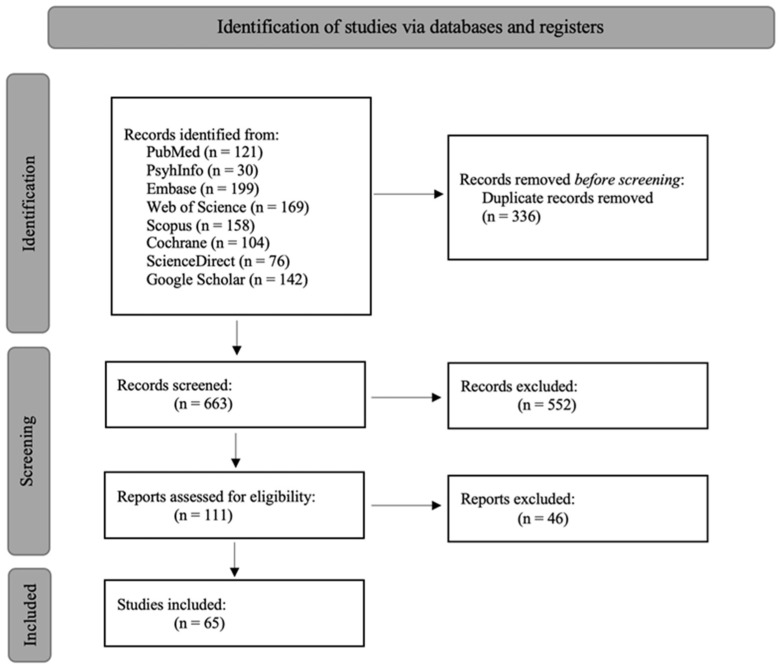
Flow diagram of study selection.

**Figure 2 healthcare-12-01333-f002:**
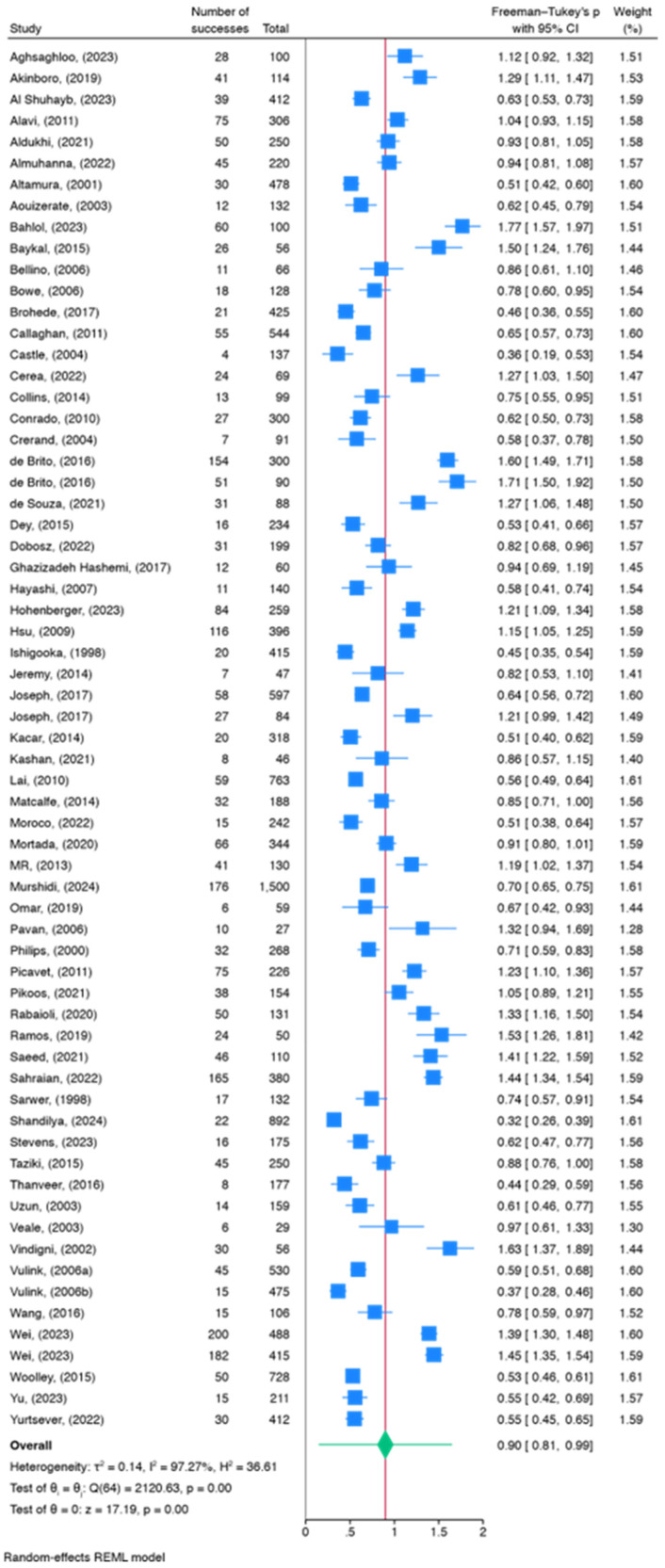
Forest Plot of Meta-Analysis. The square point of each line represents the BDD prevalence of each study. The diamond at the bottom represents the pooled effect estimate, with its width indicating the overall precision of the meta-analysis [18,19,20,21,22,23,24,25,26,27,28,29,30,31,32,33,34,35,36,37,38,39,40,41,42,43,44,45,46,47,48,49,50,51,52,53,54,55,56,57,58,59,60,61,62,63,64,65,66,67,68,69,70,71,72,73,74,75,76,77,78,79,80,81].

**Figure 3 healthcare-12-01333-f003:**
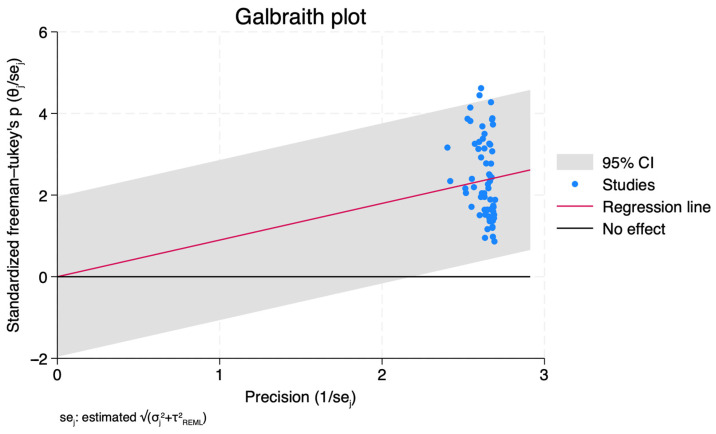
Galbraith Plot. This plot shows the relationship between study effect sizes and their precision. Points closer to the regression line indicate higher precision, suggesting more reliable estimates.

**Figure 4 healthcare-12-01333-f004:**
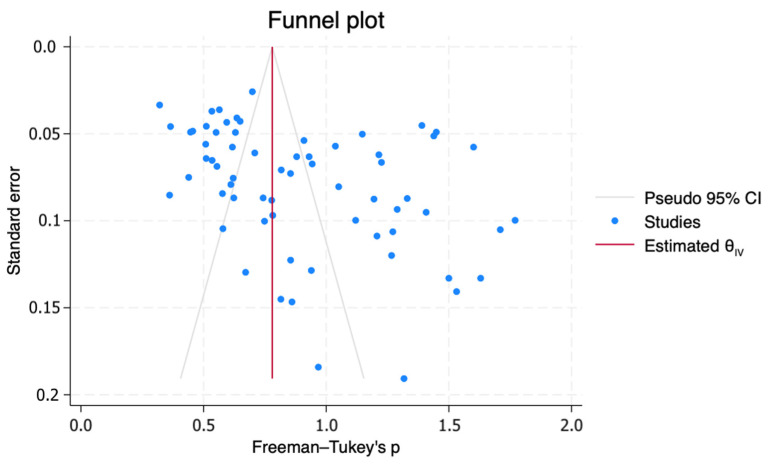
Funnel Plot. The plot shows study size versus effect size. Symmetry indicates no publication bias; asymmetry suggests potential bias.

**Table 1 healthcare-12-01333-t001:** Characteristics of meta-analysis studies with prevalence of body dysmorphic disorder (BDD). Body Dysmorphic Disorder Questionnaire (BDDQ), Body Dysmorphic Disorder Questionnaire—Dermatology Version (BDDQ-DV), Body Dysmorphic Disorder Questionnaire—Aesthetic Version (BDDQ-AS), Body Dysmorphic Symptoms (BDD-S), Body Dysmorphic Disorder Examination—Self Report (BDDE-SR), Body Dysmorphic Symptom Scale (BDSS), Body Dysmorphic Metacognition Questionnaire (BDMÇQ), Body Image Concern Inventory (BICI), Body Shape Questionnaire-16 (BSQ-16), Dysmorphic Concerns Questionnaire (DCQ), Structured Clinical Interview for DSM-IV Axis I/II Disorders (SCID-I/II), Mini International Neuropsychiatric Interview Plus (MINIPLUS), Diagnostic and Statistical Manual of Mental Disorders, fourth edition, text revision (DSM IV-TR), Self-Rating Scale of Body Image (SSBI), Body Dysmorphic Questionnaire-Aesthetic Surgery (BDDQ-AS), and Mandatory Psychiatry Evaluation (MPE).

First Author, Year ^Ref^	Sample Size Total	Population Type	Female N	Male N	Mean Age ± SD	Screening Tool	BDD Total, %	Type of Specialty	Country	Quality
Aghsaghloo, 2023 [18]	100	Rhinoplasty	68	32	29.4 ± 8.2	Y-BOCS	28.0%	Plastic Surgery	Iran	High
Akinboro, 2019 [19]	114	Dermatologic	67	47	37.0 ± 17.5	Y-BOCS	36.0%	Dermatology	Nigeria	High
Al Shuhayb, 2023 [20]	412	Dermatologic	301	111	-	BDDQ	9.5%	Dermatology	Saudi Arabia	High
Alavi, 2011 [21]	306	Rhinoplasty	245	61	23.0 ± 4.9	DSM IV-TR	24.5%	Plastic Surgery	Iran	High
Aldukhi, 2021 [22]	250	Dermatologic	224	26	33.2 ± 11.4	BDDQ-DV	20.0%	Dermatology	Saudi Arabia	High
Almuhanna, 2022 [23]	220	Aesthetics	220	0	-	Y-BOCS	20.5%	Plastic Surgery	Saudi Arabia	High
Altamura, 2001 [24]	478	Aesthetics	364	114	32.4 ± 11.5	Y-BOCS	6.3%	Plastic Surgery	Italy	High
Aouizerate, 2003 [25]	132	Aesthetics	124	8	40.6 ± 12.9	DSM IV-TR	9.1%	Plastic Surgery	France	Medium
Bahlol, 2023 [26]	100	Aesthetics	70	30	28.3 ± 8.5	BDDQ	60.0%	Plastic Surgery	Iraq	Medium
Baykal, 2015 [27]	56	Rhinoplasty	31	25	27.9	BDDQ	46.4%	Plastic Surgery	Turkey	High
Bellino, 2006 [28]	66	Aesthetics	57	9	43.4 ± 12.1	Y-BOCS	16.7%	Plastic Surgery	Italy	High
Bowe, 2006 [29]	128	Dermatologic	92	36	24.1 ± 8.3	BDDE-SR	14.1%	Dermatology	USA	High
Brohede, 2017 [30]	425	Dermatologic	425	0	39.7 ± 12.2	BDDQ	5.0%	Dermatology	Sweden	High
Callaghan, 2011 [31]	544	Aesthetics	373	171	19.3 ± 3.1	BDDQ	10.1%	Plastic Surgery	USA	High
Castle, 2004 [32]	137	Aesthetics	119	18	40.2 ± 10.4	DCQ	2.9%	Plastic Surgery	Australia	High
Cerea, 2022 [33]	69	Aesthetics	62	7	39.8 ± 14.2	BDD-S	34.8%	Plastic Surgery	Italy	Medium
Collins, 2014 [34]	99	Reconstructive	53	46	26.7	BDDE-SR	13.0%	OMFS	USA	High
Conrado, 2010 [35]	300	Dermatologic	279	21	42.2	BDDQ	9.1%	Dermatology	Brazil	Medium
Crerand, 2004 [36]	91	Aesthetics	48	43	34.6 ± 15.9	BDDE	7.7%	Plastic Surgery	USA	High
de Brito, 2016 [37]	300	Aesthetics	256	44	38.5 ± 11.3	BDDE	51.5%	Plastic Surgery	Brazil	Medium
de Brito, 2016 [38]	90	Aesthetics	84	6	38.0 ± 11.0	BDDE	57.0%	Plastic Surgery	Brazil	Medium
de Souza, 2021 [39]	88	Rhinoplasty	66	22	-	BDSS	35.1%	Plastic Surgery	Brazil	Medium
Dey, 2015 [40]	234	Aesthetics	157	77	47.0 ± 16.0	BDDQ	6.8%	Plastic Surgery	USA	High
Dobosz, 2022 [41]	199	Aesthetics	189	3	-	Custom	15.6%	Dermatology	Poland	High
GH, 2017 [42]	60	Rhinoplasty	24	36	26.7 ± 6.9	BICI	20.0%	Plastic Surgery	Iran	High
Hayashi, 2007 [43]	140	Aesthetics	124	16	38.4	DSM IV-TR	7.8%	Plastic Surgery	Japan	High
Hohenberger, 2023 [44]	259	Aesthetics	151	108	29.6 ± 16.0	BDDQ-AS	32.5%	ENT	Germany	High
Hsu, 2009 [45]	396	Dermatologic	-	-	-	Custom	29.4%	Dermatology	Singapore	High
Ishigooka, 1998 [46]	415	Aesthetics	285	130	35.0 ± 13.7	Not specified	4.8%	Plastic Surgery	Japan	High
Jeremy, 2014 [47]	47	Rhinoplasty	20	27	31.1	BDDQ	15.0%	Plastic Surgery	Singapore	High
Joseph, 2017 [48]	597	Aesthetics	398	197	46.6 ± 16.3	BDDQ	9.7%	Plastic Surgery	USA	High
Joseph, 2017 [49]	84	Aesthetics	39	45	45.7 ± 18.7	BDDQ	32.0%	ENT	UK	High
Kacar, 2014 [50]	318	Dermatologic	212	106	32.9 ± 11.5	BDDE-SR	6.2%	Dermatology	Turkey	High
Kashan, 2021 [51]	46	Aesthetics	26	20	42.9	BDDQ	16.7%	OMFS	USA	Medium
Lai, 2010 [52]	763	Aesthetics	671	92	-	DSM IV-TR	7.7%	Plastic Surgery	Taiwan	High
Matcalfe, 2014 [53]	188	Reconstructive	188	0	51.0 ± 10.0	DCQ	17.0%	Plastic Surgery	USA	High
Moroco, 2022 [54]	242	General	151	91	53.7 ± 17.3	BDDQ	6.2%	ENT	USA	High
Mortada, 2020 [55]	344	Aesthetics	296	48	39.7 ± 13.8	BDDQ	19.2%	Plastic Surgery	Saudi Arabia	High
MR, 2013 [56]	130	Rhinoplasty	99	31	26.4 ± 6.3	BDDQ	31.5%	Plastic Surgery	Iran	High
Murshidi, 2024 [57]	1500	Dermatologic	1140	360	29.3 ± 14.8	DCQ	11.7%	Dermatology	Jordan	High
Omar, 2019 [58]	59	Rhinoplasty	48	11	26.4 ± 4.8	SCID-I/II	10.2%	Plastic Surgery	Egypt	Medium
Pavan, 2006 [59]	27	Aesthetics	23	4	35.2 ± 12.8	MINIPLUS	37.0%	Plastic Surgery	Italy	High
Phillips, 2000 [60]	268	Dermatologic	187	81	42.8 ± 16.0	BDDQ	11.9%	Dermatology	USA	High
Picavet, 2011 [61]	226	Rhinoplasty	124	102	33.0 ± 16.0	Y-BOCS	33.0%	ENT	Belgium	High
Pikoos, 2021 [62]	154	Aesthetics	154	0	44.9 ± 11.6	BDDQ-DV	25.0%	Plastic Surgery	Australia	High
Rabaioli, 2020 [63]	131	Rhinoplasty	78	53	36.3 ± 14.1	BDDE	38.0%	Plastic Surgery	Brazil	High
Ramos, 2019 [64]	50	Rhinoplasty	39	11	32.3 ± 11.0	Y-BOCS	48.0%	Plastic Surgery	Brazil	Medium
Saeed, 2021 [65]	110	Rhinoplasty	110	0	-	DCQ	41.8%	Plastic Surgery	Pakistan	Medium
Sahraian, 2022 [66]	380	Rhinoplasty	210	170	-	BDMÇQ	43.4%	Plastic Surgery	Iran	Medium
Sarwer, 1998 [67]	132	Aesthetics	100	32	-	BDDE-SR	12.9%	Plastic Surgery	USA	High
Shandilya, 2024 [68]	892	Rhinoplasty	-	-	-	MPE	2.5%	Plastic Surgery	Ireland	High
Stevens, 2023 [69]	175	Aesthetics	121	54	57.5	DCQ	9.1%	Plastic Surgery	USA	High
Taziki, 2015 [70]	250	Rhinoplasty	220	30	24 ± 4.7	DCQ	18.0%	Plastic Surgery	Iran	High
Thanveer, 2016 [71]	177	Dermatologic	95	82	30.5 ± 9.9	BDDQ-DV	4.5%	Dermatology	India	Medium
Uzun, 2003 [72]	159	Dermatologic	77	82	19.5 ± 3.9	SCID-I/II	8.8%	Dermatology	Turkey	High
Veale, 2003 [73]	29	Rhinoplasty	22	7	38.0 ± 12.8	Y-BOCS	20.7%	Plastic Surgery	UK	High
Vindigni, 2002 [74]	56	Aesthetics	45	11	36.3 ± 13.0	MINIPLUS	53.0%	Plastic Surgery	Italy	Medium
Vulink, 2006a [75]	530	Aesthetics	-	-	33.6 ± 14.9	BDDE-SR	8.5%	Dermatology	Netherlands	Medium
Vulink, 2006b [75]	475	Dermatologic	-	-	34 ± 14.7	BDDE-SR	3.2%	Plastic Surgery	Netherlands	Medium
Wang, 2016 [76]	106	Aesthetics	106	0	33.1 ± 12.4	BDDE	14.2%	Plastic Surgery	China	Medium
Wei, 2023 [77]	488	Aesthetics	367	121	-	BBDQ-AS	41.0%	Plastic Surgery	USA	High
Wei, 2023 [78]	415	Aesthetics	304	111	37.0	BBDQ-AS	43.9%	Plastic Surgery	USA	High
Woolley, 2015 [79]	728	General	-	-	-	DCQ	6.9%	Plastic Surgery	USA	High
Yu, 2023 [80]	211	Rhinoplasty	151	60	-	SSBI	7.3%	Plastic Surgery	China	High
Yurtsever, 2022 [81]	412	Dermatologic	397	15	35.8 ± 7.6	BSQ-16	7.3%	Dermatology	Poland	High

**Table 2 healthcare-12-01333-t002:** Results of Meta-Analysis. BDD prevalence in individuals of studies with 95% confidence intervals.

Study	Proportion %	[95% conf. Interval]	Weight
Aghsaghloo, (2023) [18]	0.280	0.196, 0.372	1.51
Akinboro, (2019) [19]	0.360	0.274, 0.451	1.53
Al Shuhayb, (2023) [20]	0.095	0.068, 0.125	1.59
Alavi, (2011) [21]	0.245	0.198, 0.295	1.58
Aldukhi, (2021) [22]	0.200	0.153, 0.252	1.58
Almuhanna, (2022) [23]	0.205	0.154, 0.261	1.57
Altamura, (2001) [24]	0.063	0.043, 0.087	1.60
Aouizerate, (2003) [25]	0.091	0.047, 0.147	1.54
Bahlol, (2023) [26]	0.600	0.502, 0.694	1.51
Baykal, (2015) [27]	0.464	0.334, 0.596	1.44
Bellino, (2006) [28]	0.167	0.085, 0.268	1.46
Bowe, (2006) [29]	0.141	0.086, 0.207	1.54
Brohede, (2017) [30]	0.050	0.031, 0.073	1.60
Callaghan, (2011) [31]	0.101	0.077, 0.128	1.60
Castle, (2004) [32]	0.029	0.006, 0.065	1.54
Cerea, (2022) [33]	0.348	0.239, 0.465	1.47
Collins, (2014) [34]	0.130	0.070, 0.204	1.51
Conrado, (2010) [35]	0.091	0.061, 0.126	1.58
Crerand, (2004) [36]	0.077	0.030, 0.142	1.50
de Brito, (2016) [37]	0.515	0.458, 0.571	1.58
de Brito, (2016) [38]	0.570	0.466, 0.671	1.50
de Souza, (2021) [39]	0.351	0.254, 0.454	1.50
Dey, (2015) [40]	0.068	0.039, 0.104	1.57
Dobosz, (2022) [41]	0.156	0.109, 0.210	1.57
Ghazizadeh Hashemi, (2017) [42]	0.200	0.107, 0.312	1.45
Hayashi, (2007) [43]	0.078	0.039, 0.129	1.54
Hohenberger, (2023) [44]	0.325	0.269, 0.383	1.58
Hsu, (2009) [45]	0.294	0.250, 0.340	1.59
Ishigooka, (1998) [46]	0.048	0.029, 0.071	1.59
Jeremy, (2014) [47]	0.150	0.060, 0.268	1.41
Joseph, (2017) [48]	0.097	0.074, 0.122	1.60
Joseph, (2017) [49]	0.320	0.224, 0.424	1.49
Kacar, (2014) [50]	0.062	0.038, 0.091	1.59
Kashan, (2021) [51]	0.167	0.071, 0.290	1.40
Lai, (2010) [52]	0.077	0.059, 0.097	1.61
Matcalfe, (2014) [53]	0.170	0.119, 0.227	1.56
Moroco, (2022) [54]	0.062	0.035, 0.096	1.57
Mortada, (2020) [55]	0.192	0.152, 0.235	1.59
MR, (2013) [56]	0.315	0.238, 0.398	1.54
Murshidi, (2024) [57]	0.117	0.101, 0.134	1.61
Omar, (2019) [58]	0.102	0.036, 0.194	1.44
Pavan, (2006) [59]	0.370	0.196, 0.562	1.28
Philips, (2000) [60]	0.119	0.083, 0.161	1.58
Picavet, (2011) [61]	0.330	0.270, 0.393	1.57
Pikoos, (2021) [62]	0.250	0.185, 0.322	1.55
Rabaioli, (2020) [63]	0.380	0.299, 0.465	1.54
Ramos, (2019) [64]	0.480	0.342, 0.620	1.42
Saeed, (2021) [65]	0.418	0.327, 0.512	1.52
Sahraian, (2022) [66]	0.434	0.384, 0.484	1.59
Sarwer, (1998) [67]	0.129	0.077, 0.192	1.54
Shandilya, (2024) [68]	0.025	0.016, 0.036	1.61
Stevens, (2023) [69]	0.091	0.052, 0.139	1.56
Taziki, (2015) [70]	0.180	0.135, 0.230	1.58
Thanveer, (2016) [71]	0.045	0.019, 0.081	1.56
Uzun, (2003) [72]	0.088	0.048, 0.138	1.55
Veale, (2003) [73]	0.207	0.076, 0.376	1.30
Vindigni, (2002) [74]	0.530	0.398, 0.660	1.44
Vulink, (2006a) [75]	0.085	0.063, 0.110	1.60
Vulink, (2006b) [75]	0.032	0.018, 0.050	1.60
Wang, (2016) [76]	0.142	0.081, 0.216	1.52
Wei, (2023) [77]	0.410	0.367, 0.454	1.60
Wei, (2023) [78]	0.439	0.392, 0.487	1.59
Woolley, (2015) [79]	0.069	0.052, 0.089	1.61
Yu, (2023) [80]	0.073	0.041, 0.112	1.57
Yurtsever, (2022) [81]	0.073	0.050, 0.100	1.59
Invftukey (theta)	0.186	0.151	0.224

Test of theta = 0: z = 17.19, Prob > |z| = 0.0000. Test of homogeneity: Q = chi2(64) = 2120.63, Prob > Q = 0.0000.

**Table 3 healthcare-12-01333-t003:** Results of Subgroup Meta-analysis.

Group	No. of			
	Studies	Proportion	[95% conf. Interval]	*p*-Value
Country				
Australia	2	0.117	0.000, 0.406	0.070
Belgium	1	0.330	0.270, 0.393	0.000
Brazil	6	0.384	0.235, 0.545	0.000
China	2	0.102	0.044, 0.179	0.000
Egypt	1	0.102	0.036, 0.194	0.000
France	1	0.091	0.047, 0.147	0.000
Germany	1	0.325	0.269, 0.383	0.000
India	1	0.045	0.019, 0.081	0.000
Iran	6	0.275	0.202, 0.355	0.000
Iraq	1	0.600	0.502, 0.694	0.000
Ireland	1	0.025	0.016, 0.036	0.000
Italy	5	0.270	0.114, 0.462	0.000
Japan	2	0.058	0.032, 0.090	0.000
Jordan	1	0.117	0.101, 0.134	0.000
Netherlands	2	0.056	0.015, 0.118	0.000
Nigeria	1	0.360	0.274, 0.451	0.000
Pakistan	1	0.418	0.327, 0.512	0.000
Poland	2	0.109	0.042, 0.204	0.000
Saudi Arabia	4	0.169	0.116, 0.229	0.000
Singapore	2	0.231	0.109, 0.379	0.000
Sweden	1	0.050	0.031, 0.073	0.000
Taiwan	1	0.077	0.059, 0.097	0.000
Turkey	3	0.172	0.012, 0.450	0.013
United Kingdom	2	0.283	0.189, 0.388	0.000
United States	15	0.140	0.092, 0.197	0.000
Type of Specialty				
Dermatology	15	0.121	0.084, 0.163	0.000
ENT	4	0.217	0.062, 0.432	0.000
OMFS	2	0.140	0.087, 0.203	0.000
Plastic Surgery	44	0.216	0.155, 0.228	0.000
Overall				
invftukey (theta)	65	0.186	0.151, 0.224	0.000

## Data Availability

All data are available upon reasonable request from the corresponding author.

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
