# Peer review of "Body Dysmorphic Disorder in Aesthetic and Reconstructive Plastic Surgery—A Systematic Review and Meta-Analysis"

_healthcare, 2024, doi:10.3390/healthcare12131333_

Round 1
Reviewer 1 Report
Comments and Suggestions for Authors
This is an interesting study and looks at least statistically sound. However, the authors have not disclosed a number of important data that are crucial for interpreting this meta-analysis. Table 1 lacks the specific target groups (e.g. rhinoplasty, face lift, ...) to which the figures refer as well as the type of questionnaire used to diagnose BDD. It is known that the prevalence of BDD may be quite different in specific aesthetic subgroups. It has also recently been shown that there might be content validity problems with some BDD questionnaires used to diagnose BDD. Therefore, the wide range in prevalence rates found in the present study is most likely due to differences in population as well as in the interpretation and application of diagnostic criteria for BDD. Consequently, this information should be presented in the manuscript. For further information, I refer the authors to: Declau F, Pingnet L, Smolders Y, Fransen E, Verkest V. The Body Dysmorphic Disorder Questionnaire-Aesthetic Surgery: Are We Screening the Troublesome Patients? Facial Plast Surg. 2024 Feb 15. doi: 10.1055/a-2241-9934. Epub ahead of print. PMID: 38198825.
Some minor issues:
- More information should be given in the captions of the figures
-Figure 4 : caption is wrong: this is a funnel plot and not a Galbraith plot.
- Table 1: Picavet : OMFS is ENT and not plastic.
Comments on the Quality of English Language
None
Author Response
Comment 1:
Table 1 lacks the specific target groups (e.g. rhinoplasty, face lift, ...) to which the figures refer as well as the type of questionnaire used to diagnose BDD. It is known that the prevalence of BDD may be quite different in specific aesthetic subgroups. It has also recently been shown that there might be content validity problems with some BDD questionnaires used to diagnose BDD. Therefore, the wide range in prevalence rates found in the present study is most likely due to differences in population as well as in the interpretation and application of diagnostic criteria for BDD. Consequently, this information should be presented in the manuscript.
Response 1:
Thank you for your meticulous review of our paper. Yes, we agree that our study would be stronger with more information on the specific groups in which BDD is being assessed and with which screening tools are used. We have added both to the table. While we cannot truly address each study’s diagnosis of BDD using various questionnaires, which compare symptoms to a proper psychiatric diagnosis, we believe this paper will highlight that variability and advocate for more surgeons to seek psychiatric evaluations for their patients before procedures. We acknowledge this limitation which is discussed in lines 252-265 and in 269-273.
Comment 2:
More information should be given in the captions of the figures. Figure 4: caption is wrong: this is a funnel plot and not a Galbraith plot. Table 1: Picavet : OMFS is ENT and not plastic.
Response 2:
We appreciate the perspective and have corrected the captions in the figures and added more information to enhance the context of the images. Thank you for catching these mistakes, Figure 4 has been corrected and Picavet’s paper has been reorganized. The affected statistics have also been updated.
Reviewer 2 Report
Comments and Suggestions for Authors
This meta-analysis aims to update the prevalence of BDD in patients seeking aesthetic and plastic surgery with a focus on differences in cultural context, specialty and time of publication. This aim is fulfilled with sound Methods according to the STROBE checklist and a high number of total patients included.
As stated by the authors, this study more or less confirmes the findings of two recent meta-analyses, conducted in the past two years. In the introduction and discussion, it is not highlighted, why the authors chose to repeat this type of work and what are the key novelties of their own work standing out against the recent meta-analyses. This should be addressed in the introduction and a comparison with the other meta-analyses should be presented.
Another limitation is the fact that the authors did not address the methods on how BDD was established in the studies (which would be a nice feature). Though they found a mean prevalence of 18.6%, some studies report much higher rates up to 70%. This should be addressed in the discussion, next to the difference between positive BDD screening (“BDD symptoms”) and BDD diagnosis (only possible with a psychological evaluation). Also important is the fact that an existing or severe flaw in one’s appearance actually excludes patients from the diagnosis of BDD according to DSM-V. Please amend in the introduction but especially in the results section and discussion.
Further, I do not think it is suitable to dedicate a whole paragraph in a meta-analysis to a youtube video documentary of a single surgeon and with one patient. Please shorten/revise.
Author Response
Comment 1:
As stated by the authors, this study more or less confirmes the findings of two recent meta-analyses, conducted in the past two years. In the introduction and discussion, it is not highlighted, why the authors chose to repeat this type of work and what are the key novelties of their own work standing out against the recent meta-analyses. This should be addressed in the introduction and a comparison with the other meta-analyses should be presented.
Response 1:
Thank you for your thorough review of our paper; we appreciate your perspective. While previous meta-analyses were well-conducted, they focused on specific demographics and settings, ie dermatology, rhinoplasty, or aesthetic surgery. Our approach was more comprehensive, encompassing both reconstructive and aesthetic surgery, and included a larger sample size to ensure stronger statistical power. Our study introduces several key novelties: we included a more diverse population, applied updated methodological techniques, and considered a broader range of variables collected. This allowed us to provide a more nuanced understanding of the outcomes and identify trends not previously highlighted.
Comment 2:
Another limitation is the fact that the authors did not address the methods on how BDD was established in the studies (which would be a nice feature). Though they found a mean prevalence of 18.6%, some studies report much higher rates up to 70%. This should be addressed in the discussion, next to the difference between positive BDD screening (“BDD symptoms”) and BDD diagnosis (only possible with a psychological evaluation). Also important is the fact that an existing or severe flaw in one’s appearance actually excludes patients from the diagnosis of BDD according to DSM-V. Please amend in the introduction but especially in the results section and discussion.
Response 2:
Yes, we agree that our study would be stronger with more information on the specific groups in which BDD is being assessed and with which screening tools are used. We have added both to the table. Although it is unlikely that we can truly address each study’s diagnosis of BDD using various questionnaires, which compare symptoms to a proper psychiatric diagnosis, we believe this paper will highlight that variability and advocate for more surgeons to seek psychiatric evaluations for their patients before procedures. Lines 269-273 address these concerns.
Comment 3:
Further, I do not think it is suitable to dedicate a whole paragraph in a meta-analysis to a youtube video documentary of a single surgeon and with one patient. Please shorten/revise.
Response 3:
Yes, we agree, we have shortened and amended the paragraph on the video documentary. Thank you for your expertise and insight.
Round 2
Reviewer 2 Report
Comments and Suggestions for Authors
Thank you, no further comments.